# What Is Missing in Offshore Explosive Ordnance Disposal Risk Assessment?

**DOI:** 10.3390/toxics12070468

**Published:** 2024-06-27

**Authors:** Torsten Frey

**Affiliations:** GEOMAR Helmholtz Centre for Ocean Research Kiel, 24148 Kiel, Germany; tfrey@geomar.de; Tel.: +49-431-600-2599

**Keywords:** unexploded ordnance (UXO), explosive ordnance disposal (EOD), risk assessment, D81

## Abstract

Offshore explosive ordnance disposal (EOD) in the marine environment is a high-risk activity. Structured risk assessment (RA) can be a helpful tool to provide EOD experts with decision-making support. This paper hypothesizes that existing RA approaches that address unexploded ordnance (UXO) in the sea do not meet the requirements of EOD RA. To test this hypothesis, the paper proposes a novel categorization tool. It uses five review criteria: study type (qualitative vs. quantitative), level of decision-making (strategic vs. applied), risk component (probability vs. consequence), spatial scale (global vs. local), and temporal scale (long-term vs. short-term). The categorization tool is used to identify the requirements of EOD RA and to test whether nine existing RA methods fulfill these requirements. The study finds that none of the investigated RAs meets the requirements and, hence, concludes that a new method should be developed. However, some aspects of the existing studies should be considered when designing a new method. This includes using risk factors (type and mass of explosive material, type and state of the fuze, and water depth) that are relevant for EOD RA. It also involves setting up a directed graph to assess the complex interdependencies between these risk factors.

## 1. Introduction

Unexploded ordnance (UXO) in the sea poses a risk for offshore activities such as construction work, dredging, or fishing [1] The continued growth of the ocean economy and the intensified maritime spatial use [2] are likely to act as drivers for an increase in UXO encounters, which make it necessary to perform explosive ordnance disposal (EOD) operations. Given the explosive properties of the objects and the dynamic environmental conditions at sea, EOD work constitutes a high-risk activity [3]. To make informed decisions, EOD experts rely on their experience, historic data, and munitions databases [4]. They investigate a number of properties of the UXO object to evaluate whether it is safe to handle, safe to transport, or unsafe [5].

Structured risk assessment (RA) approaches for EOD are absent from the academic literature, pertinent handbooks, and relevant guidelines. While a structured RA cannot replace experienced experts with an on-site impression of a UXO object and its environment, it can aid their decision-making capabilities. It produces reproducible assessment results that are independent of fatigue, repetitiveness, lack of focus, and personal mood [6]. Even though a number of RA methods address the issue of UXO in the sea (see Section 3.2), these are not used during EOD decision-making. This is in line with the general observation that risk modeling and analysis methods are missing for many types of systems [7]. This paper, therefore, hypothesizes the following:

**Hypothesis 1 (H1).** *Existing RA methods are insufficient to assess EOD risk*.

To test the hypothesis, the study investigates how RA is currently used in the field of UXO and EOD. In this way, an understanding of the methodological state of the art is gained. If the hypothesis is rejected, it is not necessary to develop a new EOD RA method. Instead, one or several of the existing RA methods can be applied or adapted. If, however, the hypothesis is accepted, this demonstrates the existence of a methodological gap. Under consideration of the findings on existing methods, this gap can be described in detail. Aspects of existing RA methods that can be used to support EOD decision-making in the future can be identified. Subsequently, a new method can be developed.

The literature on risk and RA is extensive. The prospect of being able to predict the future with limited information in the present is universally attractive to many disciplines of science, fields of business, and aspects of daily life. There exist multiple different historically developed concepts of risk (risk is an expected value, risk is the probability of an event, risk is uncertainty, risk is the potential for loss, risk is probability and consequence, risk is a consequence, risk is consequence and uncertainty, and risk is the effect of uncertainty) [8].

To avoid an excessive discussion on the definitions of “risk” and “risk assessment”, it is helpful to use a basic concept of risk. To analyze risk, it is necessary to understand (1) what can happen (i.e., what can go wrong), (2) how likely it is that this will happen, and (3) if it does happen, what the consequences are [9]. The expected answers to these questions are a scenario identification, a probability of the scenario, and a measure of damage, respectively. Before attempting to address the second and third points, it must first be clear what exactly the subject of investigation is [10]. This is usually an initiating undesired event. It may be initiated by a risk source—an element that leads to risk, either by itself or in combination with other factors [11]. Risk sources must be distinguished from hazards, i.e., the property of something that has the potential to cause adverse effects [12]. On the path towards the undesired event, preventive barriers may exist. They reduce the probability of the undesired event’s occurrence. If the event occurs nonetheless, mitigation barriers may be in place that reduce the resulting consequences [10,13]. The event and the barriers are influenced by risk factors [10]. Another important concept is the risk receptor, i.e., a natural, physical, or socio-economic value that is potentially exposed to risk [14]. Once all these aspects are identified, the assessment of the probability and potential consequences of the undesired event can begin.

Note that the terms “risk assessment” and “risk analysis” are used interchangeably by the developers of the three points above [9]. This conflicts, for example, with the definition in ISO 31000, Ref. [11] according to which the risk analysis is a part of the risk assessment. Other scholars consider the risk assessment to be part of the risk analysis [10]. The finding that in the field of risk, different terms are used in different ways has been presented repeatedly (e.g., [8,15]). Apart from such dissent, the literature agrees that RA must be conducted in a structured and systematic manner.

In this study, the author proposes a new way to categorize RA methods. Five review criteria (RCs) are defined, each of which is expressed by a pair of opposing poles. They are the study type (*qualitative* vs. *quantitative*), the level of decision-making (*strategic* vs. *applied*), the risk component (*probability* vs. *consequence*), the spatial scale (*global* vs. *local*), and the temporal scale (*long-term* vs. *short-term*). Once identified, the RCs are used to define the requirements for EOD RA and to categorize nine RAs that were found in the literature. The paper concludes that existing methods are insufficient to fulfill the requirements of EOD RA. Hence, the hypothesis is accepted. To close the methodological gap, the development of a new RA approach is necessary. Nevertheless, some aspects of existing RAs can be adapted for future use (see Section 4.1).

This paper is the second in a series on EOD risk. The first paper identified and described UXO and environmental risk factors impacting EOD operations in German waters [6]. It serves as a starting point to fill the gap that is identified here. Future papers will focus on the development, testing, and application of a new EOD RA model [16] that fills the identified methodological gap.

## 2. Materials and Methods

### 2.1. Description of Review Criteria

To understand the specific problems that are addressed by RA methods, it is necessary to define criteria for their review. For this purpose, the theoretical literature that describes different criteria of RA methods was consulted. No structured literature review was conducted. Instead, textbooks and meta-analyses were used. Individual RA studies were included if they provided additional insights. Based on the literature, this section defines five RCs to categorize RA methods.

**RC1—Study Type:** Like in many other fields, one distinction between methods is between qualitative, semi-quantitative, and quantitative ones. When trying to express a given situation in a model, reality can be reconstructed qualitatively or quantitatively (or in a combination of the two). Either statements can be made to qualitatively describe the situation and, thus, the risk associated with it, or it can be modeled numerically and thus quantified [17]. Qualitative methods resort to measuring probability, consequences, and therefore risk in linguistic ways, for example, by classifying them as “high” or “low”. Sometimes indexing methods are used to produce a risk value for a qualitative RA (e.g., [18]). Quantitative analyses, on the other hand, use numerical modeling and simulations to assess risk [19]. Semi-quantitative approaches use numerical values on rating scales [20]. In practice, this often means that semi-quantitative methods only allow expressing risk as distinct values, while quantitative methods permit continuous ones.Quantitative and qualitative RA methods are suitable for achieving different goals. Comparative studies of qualitative and quantitative approaches have found that their results can be similar, so that the methodological choice can be based on the available data [18], or display significant differences [21]. In some cases, applying a combination of both types of methods may be the best option. It is also possible to use the results of qualitative RA as inputs for a quantitative one [19,22]. It can be argued that no fully quantitative RAs exist. This is due to the assumption that even though a risk value may be calculated mathematically, the assessment of that value is always conducted by an assessor who possesses a given, i.e., limited, amount of knowledge [8].RC1 describes the study type. Its spectrum ranges from *qualitative* to *quantitative* approaches. For RC1, the semi-quantitative methods are the logical midpoint between the two poles (see Table 1).**RC2—Level of Decision-Making:** RA can be used to support decision-making, from high levels of strategic consideration to detailed, applied examinations. The level of decision-making can be described as negatively correlated to the level of detail that an RA provides as output. In enterprise RA practice, a distinction between strategic and operational RA can be found [23]. In the field of UXO, RA may, for instance, be used for the prioritization of contaminated sites. However, the output of such an RA may be unsuitable to support decision-making about how to address a specific site [24]. While the prior can be considered a high-level strategic decision, the latter is more applied. This does not mean that a prioritization RA is of lower quality than an RA discussing different approaches for the management of a specific contaminated site. These are different questions, requiring different methods.RC2 expresses the level of decision-making. It distinguishes between *strategic* and *applied* RA methods. Table 1 provides examples for levels of EOD decision-making between these two poles.**RC3—Risk Component:** Historically, the use of nuclear power to produce energy [25,26] and the growing use of synthetic chemicals [27,28] drove the development of new ways to assess risk. The very different nature of these developments cultivated different RA methods. For the assessment of nuclear power plants, it is necessary to explore possible paths toward events that would lead to catastrophic consequences. Since these are considered unacceptable, RA that informs the design and construction of a nuclear power plant focuses on preventing such accidents by understanding the probability of the occurrence of certain events. In contrast, concerning the health risk of chemical exposures, it was necessary to analyze the likelihood of certain effects if a given level of exposure were to occur. Therefore, it was not only essential to be able to assess probability but also possible consequences [24]. More recently, the oil and gas sector has shown great interest in RA methodology and has made noteworthy contributions to the field [15]. Among them is the bow-tie method, which considers both the probability and the consequences of the occurrence of undesired events [29].To use its results, it is important to know to what degree an RA method addresses probability and consequences. If a bias towards either side is introduced unknowingly, the derived risk management decisions can be inappropriate. For example, if a focus is placed on assessing consequences, decisions may be overly risk-averse, even though probabilities are extremely low [30]. An informed decision to lean toward either of the two risk components is not problematic if the implications are known. Focusing on either of the RA components may also result from an uneven distribution of knowledge. Consider a situation in which the mechanisms determining the likelihood of an event are well known but data on its consequences are limited. This may lead to a tendency to use a method focusing on probability [31]. The challenge could also be addressed by assessing the probability aspect quantitatively and the consequence aspect qualitatively (see RC1).RC3 describes how strongly an RA leans towards either of the two RA aspects *probability* or *consequence*. A study that is placed at the center of the two aspects addresses both aspects to a similar degree.**RC4—Spatial Scale:** The spatial scale that can be addressed by an RA ranges from global to local. Spatial aspects are relevant when assessing geographical patterns and the limits of undesired events. In pollution RA, spatial considerations are commonplace. For example, the location of the risk source and the risk receptor are relevant information [32]. Furthermore, if pollution can spread over large distances, the spatial scale of an assessment should be selected accordingly. The nature of the risk source plays a role in spatial considerations as well. Dispersed non-point sources, by nature, require a spatial analysis of risk [33].RAs that address larger spatial scales sometimes consider variability in the properties of risk receptors (such as their vulnerability and resilience) [34]. The spatial and temporal aspects of RA are connected since the consequences of an event may have a different spatial extent [35] or result in different exposure levels for risk receptors [34] at two different points in time. Furthermore, a focus on larger geographical areas requires a method that acknowledges both spatial and temporal heterogeneity of risk factors [36]. There also exists a connection between the spatial scale and the decision-making level (RC2). A prioritization RA of different sites will, by nature, have a larger spatial scale than a site-specific RA.RC4 describes the spatial scale of the RA method. One side refers to *large* global studies. The opposite side refers to very *small* local assessments that deal with a single object and its immediate geographic vicinity. The middle ground roughly refers to an area, the size of an individual bay, or an offshore infrastructure development site, such as a wind park. Such an area will vary from project to project, but the scale in Table 1 should suffice to provide a general understanding of the categorization.**RC5—Temporal Scale:** Time is a universal aspect of risk [37]. It is usually uncertain when a risk will materialize and when its consequences will ensue. Numerous aspects of risk may change over time. Among them are the exposure to risk [38], the condition, and the vulnerability of a potentially affected risk receptor. The frequency of the occurrence of risk events is another important temporal aspect of RA [32]. No steady state of the relevant systems should be assumed, and, thus, the risk of the seemingly same event may vary significantly over time [35]. Consequentially, an RA that is executed one day may be faulty or misleading another day. The longevity of the results of an RA may vary depending on its temporal scale.The predictive capacity of an RA is the amount of time that the method is targeting to predict the probability and resulting consequences. When discussing a natural disaster, one may seek to predict the risk for many decades into the future, which would be a long-term RA. On the other hand, when assessing the accident risk of a given EOD operation, one would be interested in the very immediate probability and consequences.RC5 describes the temporal scale of the predictive capacity of an RA on a range from *long-term* to *short-term*. RC5 does not refer to time as an input parameter for risk factors. For example, the fact that some UXO objects have been submerged since 1914 while others were dumped as late as 1949 is one of many input parameters when assessing EOD risk, but it is irrelevant for RC5. 

The RCs demonstrate the diversity of RA methods. To evaluate their suitability for EOD RA, each of the five RCs is subdivided into five categories. Table 1 gives an overview of the RCs and their categories. The names of the RCs are shown in column 1. The second and last columns contain the two opposing poles of each RC. For example, for RC4, these are a large and small spatial scale. The remaining columns contain five categories per RC. In theory, this allows for 3125 combinations of categories. While many of these may not be realistic methodological setups, the number demonstrates how diverse the field of RA is. Note that the categories of RC1, RC3, and RC5 can be used to analyze RA methods on any subject matter. The categories of RC2 and RC4, on the other hand, address the field of EOD RA. If the table is applied to other subjects, these categories require changing.

### 2.2. Use of Review Criteria

With the RCs that were presented in Section 2.1, it is possible to categorize potentially needed RA methods. To do so, a category from Table 1 must be selected for each RC. In this study, this was done to define the requirements for an EOD RA method.

It is also possible to categorize existing RAs. In the case of this paper, nine RAs were identified and reviewed. They are henceforth consecutively numbered RA1 through RA9. These nine RAs were not identified by conducting a systematic literature review. As mentioned in Section 1, articles on RA concerning UXO or EOD are very much absent from the academic literature. Only two of the reviewed studies (RA6 and RA7) were published in indexed journals. Most other RAs were industry reports that were either found through exploratory search (RA1, RA3, RA5, RA8) or handed directly to the author (RA9). RA2 was a student paper, and RA4 was a doctoral thesis.

For each RA, the description of the method as well as its application were investigated to judge what kind of study was performed. Subsequently, Table 1 was again used to perform the categorization of the RCs. A categorization tool is presented that can be used to visualize which criteria an EOD RA should meet and to analyze whether existing RA methods fulfill these criteria. It is also possible to assess the degree of deviation between an RA and the requirements. Details on this comparison are given in Section 4.

## 3. Results

Section 2 described a new way of classifying RA methods with the help of five RCs, each of which is subdivided into five classes. This categorization tool is now used to identify the requirements of the EOD RA (Section 3.1) and analyze whether existing methods fulfill these requirements (Section 3.2).

### 3.1. Requirements of the Explosive Ordnance Disposal Risk Assessment

Before the requirements of the EOD RA can be formulated, it is necessary to define various other aspects of EOD risk (see Section 1). First of all, it should be noted that the UXO object that requires handling is the risk source. The explosive properties of this object and the dynamic environmental conditions constitute a hazard. Thus, EOD is considered a risky activity, and the personnel and equipment at the EOD location are the risk receptors. The occurrence of an undesired detonation was defined as the undesired event. Other undesired events, such as leakage of or direct physical contact with explosive materials, may take place as well [39]. However, they are of no concern to the risk receptors. Note that it is also possible to execute a planned detonation as part of the EOD procedure [40]. In that case, all personnel and equipment are located outside a safety perimeter. It is therefore not relevant here. At this point, it is not necessary to assess the preventive and mitigation barriers, as this would already be part of the RA itself. Now that all these aspects have been defined, the requirements of EOD RA are discussed and categorized for each RC.

**RC1—Study Type:** On the one hand, EOD is a very common activity in German waters, and data on these EOD activities exist (e.g., [41]). Such data include various UXO properties as well as environmental conditions [4]. At the same time, the accident rate in EOD is very low, which means that data on the occurrence of the undesired event (the undesired detonation) are virtually nonexistent. EOD experts execute a qualitative assessment every time they judge whether UXO is safe to handle, safe to transport, or neither [40]. Expert knowledge makes the use of rating scales feasible in cases where no data exist. In summary, it should be possible to design an EOD RA that leans towards the quantitative side of RC1. However, due to the lack of data on accidents, it must resort to expert judgment where necessary. Therefore, category 4, *rather quantitative*, is selected.**RC2—Level of Decision-Making:** The RA must focus on the risk during the execution of EOD. Note that in the literature, EOD may also refer to the entire procedure for managing UXO. This includes desk-based research, technical investigations, clearance, and disposal [40]. EOD RA should focus on UXO handling proper, that is, the three processes of underwater transfer, in-situ destruction, and recovery [6]. For this purpose, it is necessary to analyze the particularities of the given UXO object and the surrounding environment. The EOD RA is not meant to be a tool for the prioritization of dump sites or the planning of offshore construction projects in light of managing UXO. The former is currently under development in Germany [42], and the latter already exists, as will be demonstrated in Section 3.2. Hence, a method for applied decision-making is required. Therefore, category 5, *object handling*, is selected.**RC3—Risk Component:** The consequences of unintended detonations may be dire [1]. During EOD, personnel or valuable technical equipment are in the vicinity of the UXO. This circumstance drove the definition of the risk receptor. The loss of life or valuable equipment is considered unacceptable and must be avoided. Consequently, the EOD RA method should focus on the probability component of risk. However, before a better understanding of the consequences has been gained, it would be unwise to completely ignore them. Therefore, category 2, (*mainly probability*) is selected.**RC4—Spatial Scale:** EOD is a spatially limited activity. The spatial extent includes the UXO object, its immediate environment, and the EOD personnel and equipment on site. While the surrounding environment certainly influences EOD through macro-scale natural processes such as regional currents and weather conditions, the relevant conditions during EOD are those on site. Accordingly, the EOD RA deals with a small spatial scale. Therefore, category 5, *UXO object*, is selected.**RC5—Temporal Scale:** There are two dimensions to the temporal scale of the EOD RA. Firstly, it should be possible to assess the risk related to EOD right before its execution. Secondly, the occurrence of an undesired detonation is likely the instant result of a certain action, and immediate consequences must be expected. For both of these dimensions, the temporal scale that needs to be covered by the EOD RA is very short. The fact that UXO objects were submerged between 1914 and 1949 is irrelevant for the definition of the temporal scale since it does not affect the required predictive capacity of the RA. Therefore, category 5, *days and hours*, is selected.

Figure 1 displays the categorization of the requirements of EOD RA. In each line, one of the five RCs is expressed by its two poles. The space in between represents one axis per line on which the RA can be placed. The black pentagons in each line show the categorized requirements of the EOD RA. This simple RA categorization clarifies the overall functionality and area of application of any RA and permits a comparative assessment of multiple RA methods. The placement of the black pentagons produces a characteristic curve when connecting them from top to bottom. It allows checking whether the categorizations of existing RAs result in identical or similar curves. If yes, this would be an indication that a suitable RA method for EOD RA already exists. If not, a new method for EOD RA should be developed.

### 3.2. Offshore Unexploded Ordnance Risk Assessment Review

This section reviews existing RA methods that address offshore UXO. For each group of methods, it briefly discusses their theoretical background. It then presents the studies that apply the methods to offshore UXO. Finally, each RA is entered in the RA categorization tool and compared to the requirements. Each subsection contains a brief summary, which is picked up in the broader discussion in Section 4.

#### 3.2.1. Risk Matrices

A risk matrix is a tool to visually display the risk of an event by classifying the probability of its occurrence and the consequences were it to occur. Any number of categories, both for the probability and the consequences, can be chosen. It is accordingly necessary to assess both risk components for each event under observation. The display of risk in the form of a matrix is not necessarily numerical and can merely serve the purpose of visualization. Risk matrices can lead to difficulties in distinguishing between the risk of different events if the number of categories for probability and consequences is low, thereby leading to a crude result. They are, however, a useful tool for getting a rough overview of risk. The theoretical literature states that many studies multiply probability and consequence values into a risk value, which is error-prone. A risk matrix is a tool for describing risk, not for performing RA. Numerical risk values that are generated in a risk matrix strongly depend on the design of categories for consequences and probability [10].


**RA1**


A risk matrix was generated for a UXO RA study that was performed in preparation for the construction of an offshore wind park (RA1). The aim of RA1 was to assess the risk of UXO detonation during activities that had to be performed during wind park construction. The risk matrix consisted of five probability categories and five consequence categories. Each category of both components was assigned a value from 1 through 5, with 1 representing the lowest and 5 the highest value. A descriptive text for each category for both components was provided. In the matrix, the values of probability and consequences were multiplied, leading to possible risk values ranging from 1 to 25. These risk values were then again subdivided into five categories: low (1–3), low/medium (4 to 6), medium (8 to 9), medium/high (10 to 12), and high (15–25). To assess risk, the study uses various inputs. It distinguishes between four UXO types that were identified as being the most likely to occur in the area of interest of the construction project. Furthermore, it introduces eight work activities that take place during wind park construction. The study assesses risk for three risk receptors (personnel, equipment, and project progress) in two water depth classes [43]. The study appears to be strongly influenced by one of the most comprehensive UXO risk management publications available [44].

The results of RA1 are not concerned with EOD activities but with the risk of UXO to construction work. The report notes that the study was performed after the technical UXO survey [43]. When applying the RA categorization tool, RA1 is located at the center of the axis for three out of five RCs. Figure 2 displays it as a blue square. RA1 is a semi-quantitative study (RC1) that evaluates both the probability and consequences of UXO risk. Its focus is slightly shifted towards the probability aspect of risk, which is a function of the probability of encounter and the probability of detonation of a UXO object (RC3). As a planning effort for wind park development, it is also placed at the center of strategic and applied decision-making (RC2) that takes place in a mid-term time frame (RC5). Finally, the size of a wind park is the spatial scale of the study, which is a development project (RC4).


**RA2**


A risk matrix was also used in a literature review study of UXO risk (RA2). In the review, the authors identified 21 ‘UXO risk events’ and determined their likelihood of occurrence and consequence severity. Of these events, most address UXO in general, and only two are directly linked to EOD. RA2 is rather broad in its scope and addresses the overall risk of UXO. The likelihood of occurrence values were derived from the literature’s descriptions of how often the events had been reported in the past. The study distinguishes eight categories for the likelihood of occurrence, ranging from 1 to 8. For numerous risk events, the frequency could only be assumed. The study accounts for this fact by conservatively sorting these events into categories with higher occurrence values. For consequence severity, the study introduces four categories that are not mutually exclusive but may occur simultaneously. The categories were ranked 1 (property damage) through 4 (injuries, exposure, or death), and if relevant for the same event, were added up so that a maximum score of 10 could be reached. It is not fully transparent how the authors determined which of the consequence categories apply to which event. It is stated that the worst-case scenario for each event was the basis for the determination of the consequence categories. Based on the scores for the likelihood of occurrence and the consequence severity, all events were sorted into a risk matrix. In contrast to RA1, RA2 does not multiply the scores to produce a risk value. Instead, it sorts events into one of six priorities based on their location in the risk matrix [45].

RA2 (green square in Figure 2) is rated as being more qualitative than RA1, which derives its probability ratings from a historic study and intrusion depths. RA2, on the other hand, derives probability by counting how often UXO risk events are mentioned in the literature (RC1). The fact that the paper is an overall review of UXO risk means that it is of interest for answering strategic questions on a global level (RC2, RC4). It is similarly concerned with the probability and consequences of its UXO risk events and thus in the center of RC3. Finally, in terms of temporal scope, it depends on the UXO risk event. RA2 was thus placed at the center of RC5, even though individual events may warrant a different temporal scale.


**Summary on Risk Matrices**


Figure 2 shows RA1 and RA2 (green square) as profiles in the RA categorization tool. To ensure comparability, the profile of the required EOD RA (black pentagons) is also displayed. Figure 2 clearly shows how strongly the profiles of RA1 and RA2 deviate from the requirements that were formulated in Section 3.1. Hence, neither of them fulfills the requirements for this study.

#### 3.2.2. Other Semi-Quantitative Approaches

RA1 was characterized as a semi-quantitative study. Here, two additional semi-quantitative RAs are discussed. They use distinct values on rating scales to express risk. However, neither of them uses a risk matrix to express or display results.


**RA3**


A UXO risk and threat assessment that was conducted during the planning of another offshore windpark is similar to RA1. The main reason it is included in this study and described in this subsection is that it does not apply a risk matrix. RA3 investigates the probabilities of the encounter and initiation of UXO as well as the consequences of a detonation for five different types of UXO and eight types of operations that take place during the wind farm’s development. Both probability and consequences are rated from 1 (very low) to 5 (very high). They were multiplied to give a risk rating. For the probability rating, an informed judgment based on a historical desk survey was made. The consequence rating was deduced from the UXO object’s mass of explosive materials by grouping its possible size into five categories and estimating the expected consequences for four receptors (human health, plant and equipment, vessels, and environment). The method allows for a comparative description of the risk that is associated with different offshore construction operations [46].

Since RA3 is very similar to RA1, it is categorized in the same manner (blue circle in Figure 3. This means that for RC1, RC2, and RC5, nodes are located at the center of the axes. For RC3, the RA is leaning towards probability since this risk aspect is stressed more in the method than consequences. The study focuses on a development project and is thus sorted into category 4 of RC4. Even though it is more specific and transparent on the nature of the consequence rating than RA1, Figure 3 shows that RA3 is not suitable for assessing the key processes.


**RA4**


At the time of its availability, the First Risk Assessment Tool for Subaquatic Ammunition Dumpsites (FRATSAD) offered a semi-quantitative approach to assess the hazard and risk of UXO (RA4). Currently, a description of the tool can be accessed [47]. For the tool, roughly thirty conventional UXO compounds and chemical warfare agents (CWAs) were ranked according to their level of hazard. Each compound received a score in the range 1 to 15. This score was developed by combining twelve properties of the compounds in an undisclosed manner. The included properties were solubility, hydrolysis, stability, acute human toxicity, chronic human toxicity, carcinogenicity, mutagenicity, teratogenicity, aquatic toxicity, toxic hydrolysis products, bioaccumulation, and arsenic content. The tool asked the user to provide information on the amount of each compound and the size of the area in which it is distributed. This way, it calculated the concentration of each substance and the combined amount of all substances in the area. In the second step, the hazard scores of the substances were combined with the concentrations to generate a hazard value for an area. Finally, the user was required to add information on the surrounding environment. The requested information is water depth, prevalent sedimentation processes, current velocity, whether the area is part of a nature reserve, whether it is used for commercial fishing, and whether other restrictions for use exist, as well as the spatial distances to the closest waterway, the closest coastline, and the closest constructed structure. For the environmental information, underlying numerical values were used to express how they contribute to the hazard level in the area [47].

Figure 3 displays RA4 as a green circle. While working with discrete values on rating scales and therefore constituting a semi-quantitative assessment, RA4 derives its risk value from numerous measurable properties of CWAs and conventional UXO compounds. It can therefore be categorized as rather quantitative (RC1). While dealing with UXO compounds and CWAs, the results provided by RA4 are not targeted at EOD. By its developer, RA4 is considered a tool for preliminary assessment. It addresses the general public, providing a rough classification of the hazard in an area that is contaminated with UXO or packaged CWA. Hence, it is rather concerned with strategic decision-making (RC2). RA4 focuses heavily on potential consequences and does not assess the probability of them occurring (RC3). Since it is meant to be applied to any given sub-area of the Baltic Sea, the node is placed at the center of the spatial scale (RC4). Finally, the tool was designed for long-term decision-making and not for immediate planning of offshore construction or EOD work (RC5). Similar to previous RAs, the profile of RA4 deviates substantially from the requirements of the EOD RA. Another problem lies in the fact that the methodological algorithm of RA4 is undisclosed.


**Summary on Other Semi-Quantitative Approaches**


RA3 (blue circle) and RA4 (green circle) were entered into the RA categorization tool, and the resulting profiles are shown in Figure 3. Neither of them fulfills the criteria that were identified for the EOD RA (black pentagon). RA4 overlaps with the EOD due to its quantitative nature. However, its strong tendency towards assessing the consequences side of RC3 means that many of the underlying input data (e.g., on carcenogenicity, mutagenity, etc. of substances) are irrelevant for the EOD RA.

#### 3.2.3. Event Trees

Event trees can be used to assess the consequences of a given undesired event. Their purpose is to identify and quantify scenarios that may play out as an effect of the undesired event in a structured manner. To this end, a series of polar questions are asked [10]. Each of the polar questions represents an additional event that can lead to one of two possible outcomes. As the event tree forks at each of these events, potential scenarios develop, and many possible outcomes of the undesired event are determined. Event trees can be used qualitatively to produce an overview of possible scenarios that may unfold due to an undesired event. They can also be used quantitatively by assigning probabilities to the answers to each of the polar questions and, thus, to each scenario [10,48].


**RA5**


RA5 uses event trees to assess numerous pipeline construction risks (RA5). It mistakenly labels its diagrams as “fault trees”. It uses a series of percentages and probabilities to create the event tree. It does so, inter alia, to assess the risk of fatalities that may occur due to an unintentional UXO detonation during different pipeline installation operations (anchor handling, rock placement, and landfall operations). One factor that was considered in the study was the UXO density. It was derived from a technical survey that had been performed in preparation for another pipeline project. Secondly, the water depth along the pipeline route was included, and depth classes were defined. Thirdly, numerous probabilities were determined: the probability of contact with UXO during the pipeline installation operations, of the detonation of the UXO object, of encountering certain explosive charge sizes, of the sinking of a vessel, and of fatality. The values for the last one were determined by reviewing the distribution of four different weather conditions in the geographic region and the success rates of rescuing personnel under each of these conditions. The categories in the event tree (e.g., for the charge sizes) are very simplified. This may be due to the large scope of the pipeline report [49].

The event tree method, as applied in RA5, with its series of data-driven percentages and probabilities, is a rather quantitative method (RC1). As it is linked to a specific pipeline project, the decision-making level is considered to be in the middle between strategic and applied (RC2). Through its event tree approach, RA5 emphasizes probability more than consequences. It only distinguishes between the consequences of fatality and no fatality (RC3). Similar to other studies that were investigated above, RA5’s spatial scale is that of a development project (RC4). Finally, it deals with explosion risk on a medium-term temporal scale (RC5).


**Summary on Event Trees**


Figure 4 juxtaposes the profile of RA5 (blue triangle) and the requirements of the new method (black pentagon). Again, the profiles do not overlap, and thus RA5 cannot be used to perform EOD RA. However, the shape of RA5 is akin to the EOD RA’s profile.

#### 3.2.4. Risk Assessments Addressing Human or Faunal Health

The following two RAs do not use the same methodology. They are both addressed in this subsection since they address human and faunal health. While the first one is a toxicological RA, the second one addresses the risk of noise emissions during detonations.


**RA6**


In one study, toxicological health RA was performed to examine the risks of in-situ destruction (RA6). For this purpose, blue mussels were placed in the vicinity of corroded mines, free-lying hexanite, and an area with multiple pieces of exposed explosive material. After the exposure, the mussels were collected, and their tissue was analyzed for different explosive compounds [50]. Based on the data that were obtained in this analysis, the margin of exposure of the compounds for human seafood consumers was calculated using the Integrated Risk Information System of the US Environmental Protection Agency. In addition to the concentrations of the compounds in mussel tissues, input information for the calculation was the consumption of fish or seafood per capita in Germany as well as the carcinogenicity values of TNT [50].

This study’s approach differs significantly from the literature presented thus far. It addresses the potential long-term effects of the in-situ detonation during EOD. Even though the data were collected at a spatially confined dump site, the results of RA6 are universally valid and not meant to inform individual EOD campaigns. The method would be impractical for integration into the EOD framework and is thus unsuitable for the EOD RA.

Figure 5 displays the categorization of RA6 as a blue diamond. It is very data-driven and thus a largely quantitative study (RC1). The assessment may support policymakers or authorities in the management of munitions dump sites and is therefore rather strategic (RC2). Through its quantitative approach, it addresses both the probability and consequence aspects of human health risk (RC3). The spatial scale of a dump site is rather local (RC4), and the consequences it addresses are long-term (RC5).


**RA7**


Another paper uses aspects of RA to assess the impact of detonations on the hearing of harbor porpoises (RA7). Both data on detonation and the seasonal distribution of harbor porpoises were obtained for this study. Impact areas for individual detonations were modeled based on the mass of the explosive material in the UXO objects. Existing risk thresholds for sound exposure levels and resulting injuries of porpoises were used. The thresholds were combined with the modeled impact areas of the sound waves and the animals’ density to calculate the expected number of injuries occurring as a consequence of the detonations [51].

While applying a very different type of model, RA7 has a lot in common with RA6 in terms of its methodological orientation (see the green diamond in Figure 5). It is quantitative, but it also translates some of its consequences into semi-quantitative categories (RC1). As it is universally valid, it addresses a rather strategic question (RC2). Furthermore, it deals both with probability and consequences (RC3) on a rather confined spatial scale, that is, the impact radius of the detonation (RC4). However, it differs in terms of temporal scale as it assesses short-term probabilities and consequences (RC5).


**Summary on RAs Addressing Human and Faunal Health**


In Figure 5, both RA6 (blue diamond) and RA7 (green diamond) are plotted. Since they overlap in numerous RCs, the profile of RA6 is only partially visible. Given that the two RAs address completely different issues, it is remarkable how strongly they overlap in the categorization tool. This demonstrates that methods that address different issues can still pursue their aim by adhering to very similar principles. Nevertheless, neither of the methods strongly overlaps with the EOD RA requirements. RA7 shows similarities in terms of RC1 and RC5. This is related to the fact that it quantitatively addresses the short-term consequences of a detonation. This is also a requirement of an EOD RA, and hence, aspects of RA7 should be integrated there.

#### 3.2.5. Qualitative and Descriptive Approaches

As described in Section 2.1, RA may be executed in a purely qualitative way. In this case, methods such as brainstorming, structured interviews, or structured what-if-technique may be used [20]. If an assessment is methodologically robust and deals with risk in a structured manner, it qualifies as an RA. Two qualitative studies were found and analyzed.


**RA8**


RA8 was performed in preparation for potential EOD activities. These were referred to as ‘routes to impact’. It was a prerequisite for the construction of a wind park. The study assesses the risk that various EOD activities pose to cetaceans. It is primarily concerned with the potential impacts of EOD activities on the risk receptors and less with the probabilities that these consequences may arise. The routes of impact under consideration were the noise originating from UXO clearance (i.e., planned detonations), noise originating from ROVs equipped with geophysical systems, vessel noise, and the collision of cetaceans with vessels. The assessment is purely descriptive. It provides an overview of the potential impacts of each route and predicts said impact. Finally, it describes and evaluates the significance of each impact by sorting them into categories. RA8 remains ambiguous on the parameters it used for this evaluation and why certain impact routes end up in a specific impact category [52].

Of the four routes to impact presented in RA8, only one is directly connected to EOD. It describes the potential consequences of in-situ destruction of UXO. The method is therefore unsuitable for the aim of the EOD RA since it does not cover any other EOD processes, such as the underwater transfer or salvaging of UXO objects [40].

In Figure 6, RA8 is represented by a blue inverted triangle. It is, by definition, on the qualitative end of the spectrum of RC1. It is meant to cover the rather strategic question of how the wind park’s construction would affect nearby cetaceans (RC2). To do so, RA8 focuses exclusively on the consequence component of risk (RC3). Like many other RAs, it addresses the spatial scale of a development project (RC4). Its temporal scale is considered medium-term, as it discusses not only the immediate effects of in-situ detonation but also its lasting effects for cetaceans (RC5).


**RA9**


Another descriptive RA explores the UXO risk connected to another wind park construction project (RA9). It provides a detailed description of the effects of underwater detonations and formulas for safety distances that must be maintained in the event of their occurrence. All these descriptions focus on assessing the risk for a vessel at the surface, as the main concerns of the study are the life and health of the crew. The study names the likelihood of encounter, the sensitivity of fuses, the impact of construction activities on the UXO, the size of the explosive charge, the distance of the risk receptor to the UXO object, and the protective medium (such as water) as the factors that determine UXO risk. The report also briefly considers the probability of the initiation and aging of explosive material under water [53]. RA9 does not deal with the risk during EOD but describes how EOD should be executed to manage UXO risk during offshore wind park construction, which is a much wider scope.

As it is purely descriptive, RA9 is a qualitative RA (RC1). Similar to RA1, RA9 takes place in preparation of an EOD campaign and is thus located in the middle between strategic and applied levels of decision-making (RC2). It devotes a similar amount of attention to the probability and consequences of UXO risk during wind park construction (RC3). Like many other RAs, it deals with the spatial scale of a development project (RC4). It focuses on a medium-term temporal scale (RC5).


**Summary on Qualitative and Descriptive Approaches**


In familiar fashion, RA8 (blue inverted triangle) and RA9 (green inverted triangle) were mapped onto the grid of the RA categorization tool. This is shown in Figure 6. As with the majority of previous profiles, these do not even remotely correspond to the profile connecting the black pentagons, meaning that the qualitative RAs presented in this subsection do not fulfill the requirements of the EOD RA.

## 4. Discussion

The description of the results revealed that none of the nine investigated RAs can be used to conduct EOD RA. Nevertheless, this section explores which aspects of existing RAs can be used as starting points for the development of a new method. For this purpose, it first discusses the degree of overlap and deviation between the existing RAs and the requirements. For the RAs with the greatest overlap, it identifies aspects that should be considered during EOD RA (Section 4.1). Secondly, the section revisits the five RCs to offer insights on the current methodological gap (Section 4.2). Finally, it addresses some limitations of the categorization tool (Section 4.3).

### 4.1. Comparison of Existing Risk Assessments with the Requirements

The categorization of each individual RA was compared to the requirements. Table 2 shows the results. The lines contain the nine RAs, and the columns contain the five RCs. Each cell in the matrix contains a value expressing the deviation between an RA and the requirements for a specific RC. Cases where the requirements are met are represented by the deviation value 0. For example, both in RA4 and the requirements, the study type (RC1) is *rather quantitative*. The largest possible deviations are an RC’s opposing poles. They are represented by a deviation value of 4. For example, for the level of decision-making (RC2), this is the gap between RA2, which is fit to inform a *national strategy*, and the EOD RA, which is required to focus on *object handling*.

Next, the deviation values for each RC were added to a deviation score per RA. It is shown in the rightmost column of Table 2. The higher the score in this column, the greater an RA’s total deviation from the requirements. The theoretical value space of this deviation score would range from 0 (complete overlap) to 18 (maximum deviation).

On the one hand, none of the existing methods overlap fully with the formulated requirements. On the other hand, the methods do not entirely deviate from the requirements either. The deviation scores range from 4 to 13. The greater the deviation score, the lower the potential to apply the existing method to EOD. The methods of RA2, RA4, RA6, RA8, and RA9 have deviation scores of 13, 10, 8, 12, and 9, respectively. They are considered to be too methodologically removed from the requirements and are not discussed any further. The remaining RAs are revisited in the following paragraphs. Emphasis is placed on RCs for which the deviation is no greater than 1.

The comparison between RA1 and the requirements is characterized by an overlap in the risk component (RC3, *mainly probability*) and a small deviation in study type (RC1) and spatial scale (RC4). The resulting deviation score is 6. Both focus on the undesired detonation as an undesired event. However, the risk receptors for RA1 (offshore wind park construction personnel, equipment, and project progress) and the EOD RA (personnel and equipment at the EOD location) differ. In RA1, the probability of a detonation is determined by assessing various offshore activities and the probability of encountering UXO during these activities. These aspects are driven by the study’s risk receptors and cannot be used for EOD RA. Firstly, for the EOD RA, the probability of encountering UXO is always 100%. Secondly, UXO handling during EOD differs significantly from accidental encounters by non-experts. However, some other risk factors that are considered in RA1 should be incorporated into the EOD RA method. One of them is the water depth. It is used to assess the potential consequences of a detonation for unprotected personnel and equipment. The UXO type is another relevant risk factor. Numerous UXO properties depend on it, including the existence, properties, and state of the UXO object’s fuze, as well as the type and mass of the explosive material in the main charge. They control the probability of an undesired detonation independently from the risk receptor.

RA3 is similar to RA1 (see Section 3.2). It was categorized identically. Thus, its overlaps with and deviations from the requirements are identical. The same is true for the deviation score of 6.

RA5 has a deviation score of 5. It is the only existing RA that meets the requirements both for the study type (RC1, *rather quantitative*) and the risk component (RC3, *mainly probability*). A small deviation was found on the spatial scale. Its methodological approach can be a starting point for EOD RA, particularly the use of a tree-like directed graph. It allows for the complex interdependencies between numerous risk factors to be expressed. A network of UXO and environmental properties such as the one that was developed in a preceding paper [6] can be transformed into such a graph (see Section 5.2). It is akin to an event tree, with a series of probability nodes. Other important aspects of RA5 that should be considered for the EOD RA are the risk factors water depth and mass of explosive material. Both influence detonation consequences. Similar to RA1 and RA3, the probability of detonation is driven by the probability of contact with UXO and the probability that UXO detonates during offshore activities. Again, these are not useful for EOD RA.

RA7 has a deviation score of 4, which is the lowest in this study. It overlaps with the requirements for the study type (RC1, *rather quantitative*) and the temporal scale (RC5, *days and hours*). For the latter, this is the only overlap among the investigated RAs. A deviation value of 1 was found for the risk component (RC3) and the spatial scale (RC4). RA7 addresses the risk of planned in-situ detonations for harbor porpoises. Even though risk receptors and undesired events differ from the EOD RA, it is still a good starting point for understanding how the risks of detonating UXO can be assessed. It evaluates the immediate effects of detonations at different spatial scales and translates them into semi-quantitative categories. In addition, its approach to modeling impacts based on the mass of explosive material can be adapted to EOD RA.

In summary, by categorizing existing RA methods that address either UXO or EOD, it became evident that none of them fulfills the previously defined requirements. The discussion revealed that individual aspects of RA1, RA3, RA5, and RA7 should be considered for implementation into a new method. Nevertheless, a methodological gap remains, which can only be addressed by a novel approach to EOD RA.

### 4.2. Deviation of Review Criteria from the Requirements

Another series of observations can be made for the bottom line of Table 2. It shows the total number of deviation points for each RC. The minimum value in this line is 0, which would mean that all investigated RAs fulfill the requirements for an RC. The maxima depend on the categorization of the EOD RA. For RC2, RC4, and RC5, it is 36. For RC1 and RC3, it is 27, because their requirements were not categorized at either of their poles. Smaller values must be expected for the second group, and individual values must be interpreted with respect to their location in the value spaces. The following paragraphs discuss the RCs one by one.

The accumulated deviation value for RC1 (study type) is 10 out of 27. Four RAs meet the requirement *rather quantitative*. Among them are RA5 and RA7, which are methods that are closest to the requirements overall. On the other hand, RA8 and RA9 have a deviation of 3, since they are *purely qualitative*. Nevertheless, the low accumulated value of 10 signifies that existing RA methods tend to be *rather quantitative*. To inform EOD RA, existing methods can be consulted to understand how quantitative and semi-quantitative factors can be combined into one assessment.

RC2 (level of decision-making) has the highest accumulated deviation value (21 out of 36). Therefore, it is the RC for which the existing methods least satisfy the requirements. None has a deviation value of 0 or 1. While the requirements state that the EOD RA should address *object handling* by assessing the specifics of the UXO object and its environment, all existing RAs address *site management* or higher strategic levels. It can be concluded that some risk factors from existing RA methods should be considered for inclusion in EOD RA. However, they may be insufficient and lack the necessary levels of detail for a well-informed and comprehensive assessment that informs applied decision-making. Hence, a more detailed identification of risk factors should be the basis of a new method [6].

With 9 out of 27, RC3 (risk component) has the lowest accumulated deviation value. RA1, RA3, and RA5 fulfill the requirement *mainly probability*. Four other RAs deviate from the requirements by only 1 point. This demonstrates a strong tendency of the nine RA methods to focus on the probability component of risk. RA4 is the only method that focuses exclusively on consequences and therefore deviates by 3 points. For a new EOD RA method, it is therefore recommended to reuse as many risk factors as possible that determine the probability of undesired detonations (e.g., aspects relating to the UXO’s fuse properties and the explosive filling as outlined for RA1, RA3, and RA5). At the same time, numerous risk factors that define the probability were identified as offering no benefit for EOD RA (see Section 4.1). It will therefore be necessary to understand the determinants of the probability of an undesired detonation in greater detail than what is offered by existing methods. The determinants of the consequences of detonations, on the other hand (e.g., water depth and type and mass of the explosive material), are fewer and may be sufficient for an EOD RA.

Next, RC4 (spatial scale) has an accumulated deviation value of 13 out of 36. None of the existing RAs fulfills the requirement of focusing on the *UXO object* itself. Seven have a deviation value of only 1 and address risk on the scale of a *development project*. So, while a method that focuses on the individual UXO object and its surroundings is missing, some aspects of existing methods should be considered for implementation in the EOD RA. This includes the effects of detonations on risk receptors, which are addressed in RA1, RA3, RA5, and RA7. Nonetheless, it should be noted that, apart from the water depth, local environmental properties are not covered by any of the existing RAs. Understanding if and how they influence EOD risk should be a concern when developing a new method.

Finally, RC5 (temporal scale) was found to have an accumulated deviation value of 20 out of 36. Existing methods are more concerned with RA on a medium- to longer-term time scale. This finding is in line with the fact that they address higher strategic levels (RC2). The requirement of performing RA for an operation that will take place only *days and hours* into the future needs to be reflected in the EOD RA’s input data. The existing methods fail to address this requirement. The exception is the overlap of the requirements with RA7, which may be used to gather insights on the immediacy of detonation effects.

### 4.3. Limitations

If the use of the categorization tool leads to an outcome that is similar to the one presented in this paper, it is necessary to interpret the results. This calls for a good understanding of the requirements for RA in the investigated field. Without a well-thought-out set of requirements, it is not possible to understand the deviation of existing RAs and to describe the methodological gap. In addition, several of the findings that were presented in the discussion section (e.g., the expectation that determinants of the probability of an undesired detonation need to be understood in greater detail) were based on the author’s knowledge in the field of offshore UXO. If expertise in the field of application is lacking, it is therefore recommended to perform a literature review of relevant risk factors prior to or in parallel with the use of the categorization tool. After all, the review of existing RA methods can inform both studies simultaneously.

This study does not discuss how to proceed if the categorization tool reveals that a suitable RA method for a given field already exists. In such cases, some minor methodological adaptations may still be necessary. However, since the categorization was only performed with regard to EOD RA, it remains to be seen if a method that is identified as suitable can be applied without much additional effort.

Finally, it is possible that the categorization tool reveals that all investigated RAs are completely removed from the formulated requirements. Again, this study does not discuss how to proceed with such a result. In this case, the methodological gap will probably be clearly determined. However, it will be challenging to integrate other RAs’ concepts and aspects into a new method. Reviewing RAs from related fields may be a beneficial approach to gain additional insights.

## 5. Conclusions

In this conclusive section, the description and discussion of results are summarized (Section 5.1). Subsequently, potentials for their application during the development of a new EOD RA method are explored (Section 5.2). In the final subsection, the paper’s findings are considered in the wider context of what they contribute to the field of risk assessment (Section 5.3).

### 5.1. Summary of Results

As the offshore economy expands, EOD of submerged UXO grows in importance. Currently, structured RA is absent from the literature and guidelines that deal with UXO handling. Therefore, the paper presented the following hypothesis:

**Hypothesis 1 (H1).** *Existing RA methods are insufficient to assess EOD risk*.

To answer the hypothesis, the study developed and used a new categorization tool. Firstly, the requirements of the EOD RA were identified. It was found to be necessary to establish an applied, mostly quantitative method that leans towards the probability component of risk and assesses the short-term risk of EOD work on a very small spatial scale. Secondly, nine existing RAs were reviewed and categorized. None of them fulfill the requirements of the EOD RA. Hence, the hypothesis is accepted. This means that a new tool for EOD RA should be developed. While some RAs (RA1, RA3, RA5, and RA7) are methodologically close to the requirements, others deviate strongly. The former group was reviewed in greater detail to identify how they can be used as a starting point for the development of a new EOD RA method.

### 5.2. Application of Findings

Figure 7 shows the results of a study that collected relevant risk factors of UXO and the environment during an EOD operation [6]. It displays a comprehensive network of all factors that should be included in an EOD RA. The risk factors that are framed in red are addressed by the existing RAs. They include the existence, properties, and state of the UXO’s fuze to assess the probability of an undesired detonation. Furthermore, they use the mass and type of the explosive material in the main charge as well as the water depth to determine the potential consequences of a detonation. When these factors are integrated into a new EOD RA method, it is recommended to revisit how the existing studies make use of them.

The great majority of factors in Figure 7 are not considered by the nine RAs that were categorized in Section 3.2. Entire groups, such as the casing (yellow) or almost all environmental (grey) factors, have not yet been included. This underlines the methodological gap that was identified by the categorization tool.

In the discussion section, it was also suggested that the event tree that was used in RA5 can be a starting point to assess the complex interdependencies of risk factors that are relevant for EOD. While the graph in Figure 7 is not an event tree, it can be transformed into a directed graph. Here, parent nodes influence child nodes, which ultimately lead to an assessment of the probability and consequences of an undesired detonation. Such a transformation is the suggested way forward for EOD RA.

Section 4.2 identified a number of challenges for the different RCs. One challenge for EOD RA relates to RC1 (study type). It lies in the necessary transition of quantitative measures into ways to express probability and consequences. RA1 and RA3 use categories to associate certain value ranges of risk factors with certain levels of risk (e.g., a given mass of explosive material is associated with an expected level of consequence of a detonation). A similar approach may be useful for the EOD RA. Another challenge that was identified is connected to RC2 (level of decision-making), RC3 (risk component), and RC4 (spatial scale). Section 4.2 mentioned that the risk factors that were used by existing RAs are insufficient to perform a detailed RA for EOD. This issue was partially solved by Figure 7. As a next step, it will be necessary to understand the risk factors individually (i.e., their value range, how to measure them, what data are readily available), and how they are related to each other (i.e., which functions to use to calculate child nodes from parent nodes). Finally, a lack of RAs focusing on the short-term was identified for RC5 (temportal scale). Many factors in the presented network change either gradually (e.g., condition of casing, fuze corrosion state) or constantly (e.g., current velocity, wind speed). To enable a reliable EOD RA, it will be necessary to obtain high-quality and up-to-date data for both kinds of temporal effects.

### 5.3. Contributions to Risk Assessment Methodology

This paper introduces a new categorization tool for RA methods. The development and demonstration of the categorization tool are the article’s main contribution to the practice of RA. In new or less well-researched fields of application for RA, it is necessary to understand (1) the requirements of RA for the field (see Section 3.1) and (2) whether existing methods meet these requirements (see Section 3.2). If not, the categorization tool can (3) help identify which aspects of existing methods can be adapted for a new one (see Section 4.1) and which gaps must be filled (see Section 4.2). The categorization tool is, therefore, concerned with understanding the methodological requirements and the current state of the art. It is not a tool for risk analysis or management in and of itself. It may, for example, be used in a similar fashion as demonstrated above to understand the state and requirements of RA for emerging sectors (e.g., deep-sea mining or carbon capture and storage).

## Figures and Tables

**Figure 1 toxics-12-00468-f001:**
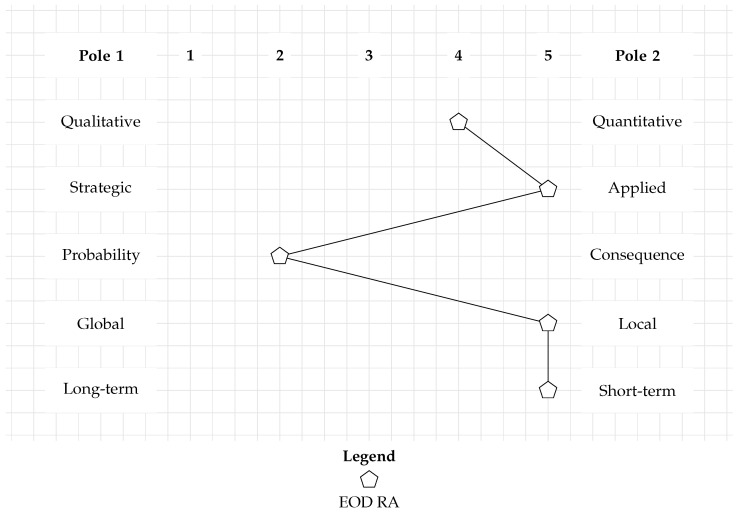
Methodological categorization of the EOD RA. All existing RA methods can be compared to this requirement profile.

**Figure 2 toxics-12-00468-f002:**
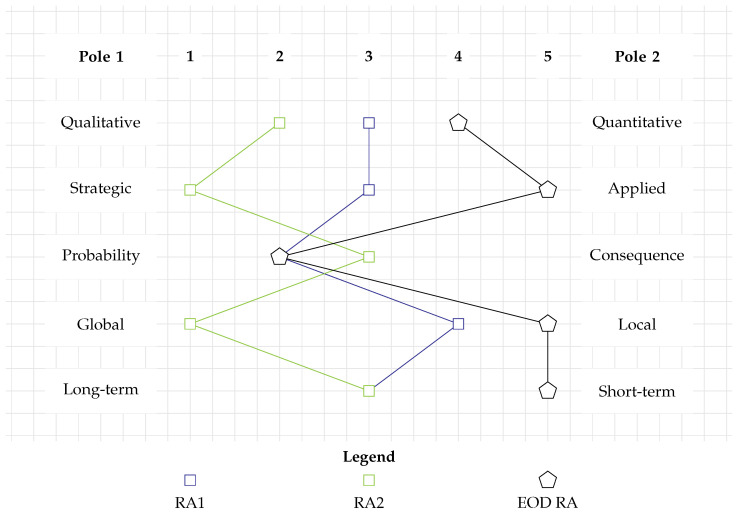
Categorization of RAs using risk matrices. Both deviate strongly from the requirements of the EOD RA.

**Figure 3 toxics-12-00468-f003:**
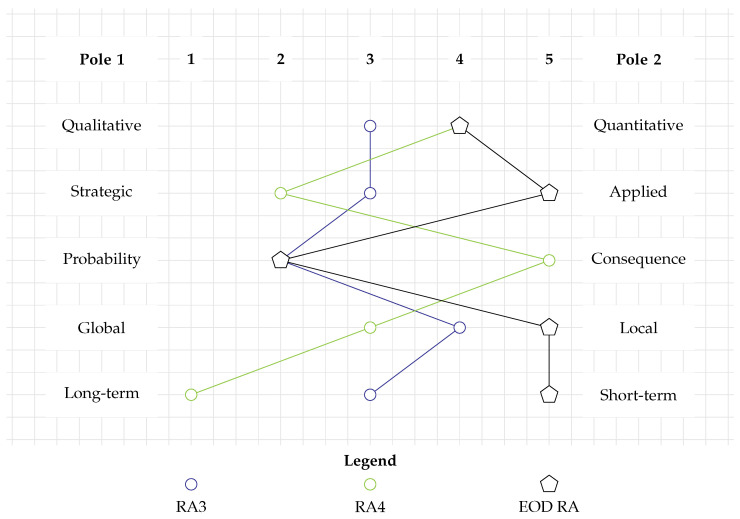
Categorization of other semi-quantitative RAs. Both deviate strongly from the requirements of the EOD RA.

**Figure 4 toxics-12-00468-f004:**
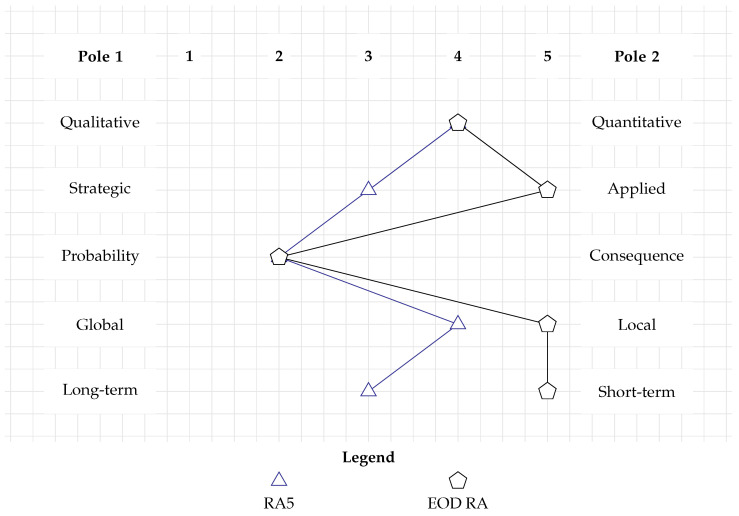
Categorization of RA5 which uses an event tree. Its profile does not overlap with the requirements of the EOD RA, but has a similar shape.

**Figure 5 toxics-12-00468-f005:**
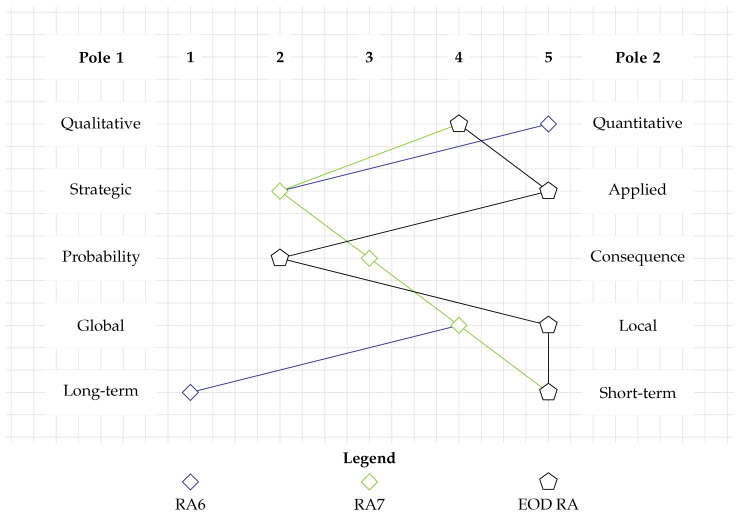
Categorization of RAs addressing human and faunal health. The profiles of RA6 and RA7 are similar to each other but do not align with the requirements for an EOD RA.

**Figure 6 toxics-12-00468-f006:**
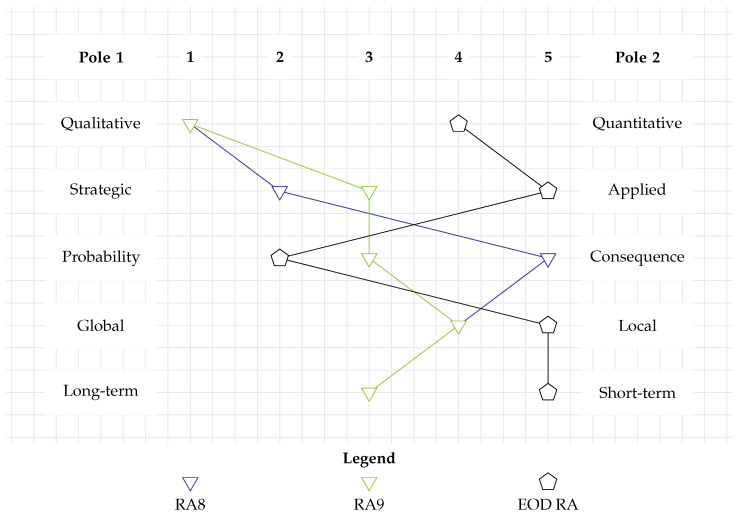
Categorization of qualitative RAs. Their profiles differ substantially from the requirements for an EOD RA.

**Figure 7 toxics-12-00468-f007:**
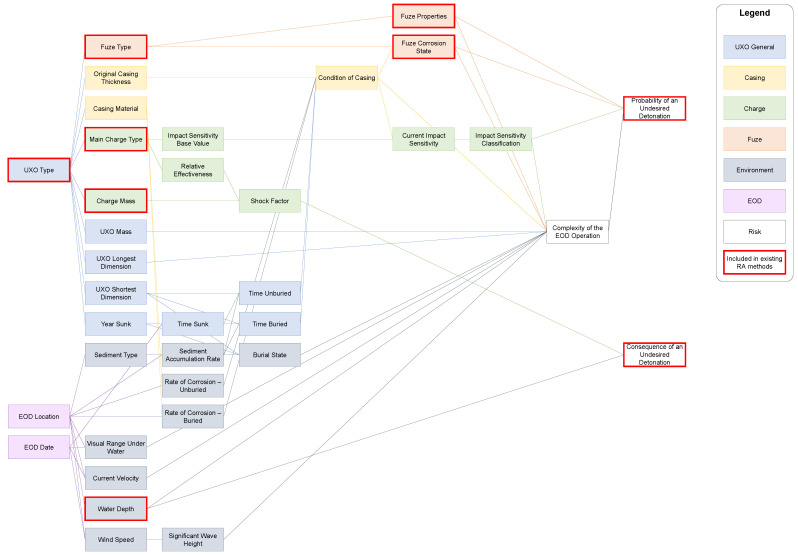
The network of UXO factors and environmental factors and their connection to EOD risk [6]. Dependencies run from left to right. Hence, a connection from one box to another signifies that the left box is the parent influencing the right box (the child). Ultimately, all connections end up influencing the probability or consequences of an undesired detonation. This network can be used as a starting point for EOD RA.

**Table 1 toxics-12-00468-t001:** Overview of RCs for the categorization of RA methods.

Review Criterion	Pole 1	Categories					Pole 2
		**1**	**2**	**3**	**4**	**5**	
RC1—Study type	Qualitative	Purely qualitative	Rather qualitative	Semi- quantitative	Rather quantitative	Purely quantitative	Quantitative
RC2—Level of decision- making	Strategic	National strategy	Site prioritization	Site management	Clearance decision	Object handling	Applied
RC3—Risk component	Probability	Solely probability	Mainly probability	Balanced	Mainly consequence	Solely consequence	Consequence
RC4—Spatial scale	Large	Global	National	Region/Bay	Development project	UXO object	Small
RC5—Temporal scale	Long-term	Decades	Years	Months	Weeks	Days and hours	Short-term

**Table 2 toxics-12-00468-t002:** Deviations between existing RAs and the requirements of the EOD RA.

Risk Assessment	Deviation					
	**RC1—Study Type**	**RC2—Level of** **Decision-** **Making**	**RC3—Risk** **Assessment** **Component**	**RC4—Spatial** **Scale**	**RC5—Temporal** **Scale**	**Deviation Score** **per RA**
RA1	1	2	0	1	2	6
RA2	2	4	1	4	2	13
RA3	1	2	0	1	2	6
RA4	0	2	2	2	4	10
RA5	0	2	0	1	2	5
RA6	0	2	1	1	4	8
RA7	0	2	1	1	0	4
RA8	3	3	3	1	2	12
RA9	3	2	1	1	2	9
All RAs	10	21	9	13	20	-

## Data Availability

The original contributions presented in the study are included in the article material; further inquiries can be directed to the corresponding author.

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
