# Peer review of "What Is Missing in Offshore Explosive Ordnance Disposal Risk Assessment?"

_toxics, 2024, doi:10.3390/toxics12070468_

Round 1

Reviewer 1 Report

Comments and Suggestions for Authors

This paper " What is Missing in Offshore Explosive Ordnance Disposal Risk Assessment?" had studied an overview of current RA practices in UXO and EOD and identify opportunities for future improvements. However, the manuscript has some defect. Thus, I suggest that the paper be Major REVISION before it is considered for publication. Thus, the authors should revise the manuscript accordingly. The following hints may help the authors:

Q1: The abstract should be improved. The abstract of the paper includes four parts: purpose, methods, results, and conclusion.

Q2: The writing and grammar should be improved. The current version of the manuscript should be considered and improved.

Q3: In general,the conclusion is not well organized. The results should be further elaborated to show how they could be used for the real applications.

Q4: The author should try to avoid the first person as much as possible, such as me, we, our and so on.

Q5: The Discussion and Results section is written too simply, and the authors are advised to rewrite the Discussion and Results section. The author should further introduce the mechanism and causes of this phenomenon.

Q6: In the introduction, the innovation of the paper and the necessity of research will not be perfectly displayed.

Comments on the Quality of English Language

Please see the Comments and Suggestions for Authors.

Author Response

Dear Reviewer,

Thank you for giving me the opportunity to submit a revised draft of my manuscript titled „What is Missing in Offshore Explosive Ordnance Disposal Risk Assessment?“ to Toxics. I appreciate the time and effort that you have dedicated to providing your valuable feedback on my manuscript. I am grateful for your insightful comments on my paper. I was able to incorporate changes to reflect the suggestions provided by you. Please find my detailed response attached.

Best regards

Reviewer 2 Report

Comments and Suggestions for Authors

 An overview of current risk assessment (RA) practices in unexploded ordnance (UXO) and offshore explosive ordnance disposal (EOD) is presented in the paper entitled “What is Missing in Offshore Explosive Ordnance Disposal Risk Assessment? The paper aims to identify opportunities for future improvements in the above risk assessment that lie in a new categorization tool. The study is relevant for the research of interest and could bring significant benefit to reduce risk when unexploded ordnance (UXO) objects are detected.  This paper could be interesting to offshore explosive ordnance disposal experts.

Indeed, the authors described various risk assessment methods that scarcely deal with UXO or EOD in the offshore environment. Indeed, the authors compare RA methods with a few methodological categorizations of the EOD RA and did not provide evidence that such a comparison is reasonable. Additionally, the explanations, that the outcomes of the RA methods described should fulfill the requirements of an EOD RA, are scarce.  

The authors state that the study is performed aiming to identify how RA can be used to better support EOD decision-making in the future. This part of the research is vaguely presented. Moreover, the need for a new EOD RA method specifically addressing UXO handling is given as a main conclusion without discussion and suggestion of what advantages of the  RA methods described could be included in it. The question ”What is Missing in Offshore Explosive Ordnance Disposal Risk Assessment?” (title of the paper) is not answered enough clearly, too.

Author Response

(The authors gave the same response as above.)

Reviewer 3 Report

Comments and Suggestions for Authors

The topic of the article is very interesting, but I have some doubts whether the work done is compatible with the scope of the journal Toxics, but I leave that judgment to the editor. However, I suggest a revision of the article in the light of the following comments.

Some of the papers cited in the literature review are relatively old, e.g. reference 2 is 43 years old, reference 6 is 25 years old, 9 is 20 years old… Given the nature of the work presented, I think the authors should present more recent work.

In line 36, the information on risk analysis is outdated; in addition to the points mentioned, prevention barriers (ability to prevent) and mitigation barriers (ability to mitigate) must be included. Furthermore, a clear distinction must be made between hazards and sources of risk. The risk assessment must be carried out systematically and take into account all aspects associated with the assessment. I suggest rewording the text to clarify this issue.

In line 53, the authors state: “In summary, there is no uniform understanding of what exactly constitutes“risk” and how it should be defined.” This statement is incorrect and should be reworded. I suggest that the author consult the ISO 31000 standard and its variants, where definitions for risk description, assessment and management can be found.

In line 58, the author states that the aim of the paper is to categorise the methods of RA and that some five review criteria have been used. With this in mind, the reader will expect that the author has applied a systematic method of literature review and conducted an exhaustive search of articles to clearly arrive at useful conclusions, however the literature review is very limited. I suggest that the review methodology is presented and the literature review is expanded, even if it is not a literature review but an article that relies only on information available in the literature; in this sense, the approach that the author must take in writing this article should be similar to that of a literature review article.

In line 114, the authors state: “Historically, the use of nuclear power for the generation of energy and the growing use of synthetic chemicals drove the development of new ways to assess risk.” Provide references to support this statement. More recently, the oil and gas industry has also contributed to the development of risk analysis and risk management methods. It can also be said that this industry is actively contributing to the development of new methods (e.g. BowTie) for risk assessment and risk management.

The author's approach to describing risk seems to me to be somewhat rudimentary, as it relies on only two components, namely impact and probability, which is indeed a somewhat outdated approach, as the risk matrix does not take into account a number of factors that are extremely important for risk assessment, e.g. to illustrate it in a simple way, travelling on a ship is risky because there is a probability that the ship will sink and the crew will drown. If we leave it at that, we have a measure of risk, but if the ship has a lifeboat, it is possible to mitigate the case of the ship sinking because the crew is saved by the lifeboat; in this sense, the risk factor is actually lower due to the presence of the lifeboat. Two conclusions can be drawn from this example: First, we must always consider the barriers of prevention and mitigation in any risk scenario. Secondly, the risk assessment and the corresponding management strongly depend on the scenario being analysed. In this sense, I propose to provide information in the article about the hazards, the sources of risk, the consequences or impacts and the barriers to risk mitigation and prevention related to the risk scenarios addressed in this study.

The author must briefly explain how this work contributes to the knowledge of risk assessment and management, i.e. in practical terms, how the information compiled in this article contributes to the practice of risk analysis and risk management.

In the conclusion, line 643, the information “This paper is the second in a series on EOD risk. The first paper identified and described UXO and environmental risk factors impacting EOD operations in German waters [1]. Future papers will focus on the development and testing of an EOD-RA model [30] that fills the gap that was identified in Section 3.4, and the application of this model.” This text should be moved to the introduction or discussion section.

In the "Discussion" section, I propose to discuss the limitations of the work carried out as well as future work.

Comments on the Quality of English Language

Minor editing of English language required

Author Response

(The authors gave the same response as above.)

Reviewer 4 Report

Comments and Suggestions for Authors

The is an interesting paper, providing an overview of current Risk Assessment (RA) methods related to Unexploded Ordnance (UXO) and Explosive Ordnance Disposal (EOD) in offshore environments. It begins by discussing the concept of risk and presents a specific definition. A categorization tool for RA methods is then introduced to define the requirements for EOD RA. Existing RA methods are reviewed, revealing that none fully meet the requirements for EOD RA.

The identified gap highlights the need for a new EOD RA method that focuses on UXO handling. This new method should be primarily quantitative, emphasizing the probability aspect and assessing short-term risks at a small spatial scale.

Overall, I have only one question, that is for such categorization, especially for the prediction of future risk, did you use any AI technique such as deep learning? I think it may be a good try and is good to your work.

Comments on the Quality of English Language

The is an interesting paper, providing an overview of current Risk Assessment (RA) methods related to Unexploded Ordnance (UXO) and Explosive Ordnance Disposal (EOD) in offshore environments. It begins by discussing the concept of risk and presents a specific definition. A categorization tool for RA methods is then introduced to define the requirements for EOD RA. Existing RA methods are reviewed, revealing that none fully meet the requirements for EOD RA.

The identified gap highlights the need for a new EOD RA method that focuses on UXO handling. This new method should be primarily quantitative, emphasizing the probability aspect and assessing short-term risks at a small spatial scale.

Overall, I have only one question, that is for such categorization, especially for the prediction of future risk, did you use any AI technique such as deep learning? I think it may be a good try and is good to your work.

Author Response

(The authors gave the same response as above.)

Round 2

Reviewer 1 Report

Comments and Suggestions for Authors

The authors have carried out a thorough and careful revision and the revised manuscript improved a lot in terms of technical quality and language. Therefore, I would recommend it for publication in the Journal.

Comments on the Quality of English Language

Please see the Comments and Suggestions for Authors.

Reviewer 2 Report

Comments and Suggestions for Authors

The author significantly improves the quality of the paper.  In the new version of the manuscript, the main problem  and the suggestion for its solving are enough clear presented.

Reviewer 3 Report

Comments and Suggestions for Authors

After analyzing the revised version of the article " What is Missing in Offshore Explosive Ordnance Disposal Risk Assessment? ", it can be stated that the authors have significantly improved the article. In this sense, I believe that the article in the revised version meets the necessary requirements to be published in the journal Toxics.

Sincerely,

Comments on the Quality of English Language

Minor editing of English language required